# TRANSFORMING OCEAN ANALYSIS: LEARNING 4D OCEAN FIELD FROM IN-SITU OBSERVATIONS VIA UNCERTAINTY-AWARE IMPLICIT REPRESENTATIONS

## ABSTRACT

A complete and accurate representation of Earth's time-evolving ocean field is crucial for understanding global warming as well as climate dynamics. However, the sparsity of current in-situ ocean measurements presents a significant challenge in estimating values in largely unobserved regions. Traditional methods, such as objective interpolation (OI), struggle with accuracy due to their reliance on discrete grids and fixed spatial correlation structures. In this paper, we propose a novel approach to reconstruct 4D ocean fields only from raw observations using implicit neural representations (INRs). Our method improves field representations by leveraging neural networks to capture continuous, complex, and nonlinear patterns inherent in ocean data. To address uncertainties in ocean measurements and the limited availability of daily observations, we incorporate uncertainty estimates and a meta-learning strategy into existing INRs. These innovations enable our approach to provide daily, resolution-free ocean temperature reconstructions, a significant improvement over monthly averaged discrete fields. Experiments demonstrate the accuracy and adaptability of our method compared with approaches, establishing our method as a transformative solution for future ocean analysis and climate monitoring.

## 1 INTRODUCTION

Global warming is one of the most crucial challenges for us. With over 90% of Earth's excess heat accumulating in the ocean, accurate ocean temperature measurements are essential to monitor this ongoing climate change. (Von Schuckmann et al., 2013; Abraham et al., 2013; Allan et al., 2014). Despite various ocean observing programs (Levitus, 1982; Argo, 2000; Goni et al., 2019; Roemmich et al., 2019), we still have a limited number of daily ocean profiling floats ($\sim$ 400 profiles). Such sparse observation impedes our understanding of global warming and its impact on climate dynamics (Lyman & Johnson, 2008; Gille, 2008; Abraham et al., 2013; Cheng & Zhu, 2014a). To overcome the shortage of raw oceanic observations, researchers develop methods to approximate or reconstruct a complete 4D ocean temperature field from in-situ observations [1] (Hosoda et al., 2008; Li et al., 2017a; Cheng et al., 2017). Known as objective interpolation (OI), these methods typically rely upon an error covariance function to correct a first-guess field (Meyers et al., 1991; Bonekamp et al., 2001). To improve the accuracy of ocean reconstruction, efforts have been made to improve both the first-guess field (Smith & Murphy, 2007; Levitus et al., 2012) and error covariance functions (Li et al., 2017a; Cheng & Zhu, 2016; Zhang et al., 2022).

However, the potential of using deep learning methods on ocean reconstruction remains under-explored. In this work, we present **MARIN**, a new AI-powered method thats integrates **M**et**A**-lea**R**ning with **I**mplicit

---

[1]In-situ observations are direct measurements of sea water variables.

Neural representation network to reconstruct complete 4D ocean data from raw oceanic observations. Unlike traditional methods that represent the first-guess field as discrete grid values, MARIN uses a neural network to model the field as a continuous function of geographical coordinates (latitude, longitude, depth). Such an idea draws inspiration from Implicit Neural Representations (INRs) (Chen et al., 2021; Sun et al., 2021; Sitzmann et al., 2020), which has shown better representation accuracy compared with discrete signal representations (Sitzmann et al., 2020; Ramasinghe & Lucey, 2022; Saragadam et al., 2023). In addition, our method corrects the first-guess field by optimizing the neural network's parameters with in-situ observations. This data-driven method addresses a major limitation of covariance functions, which are linearly modeled and often fail to account for varying spatial correlations.

Additionally, we modify and address two challenges when learning INRs to approximate ocean temperature from observation. First, ocean measurements have errors due to instruments and geolocations (Gouretski & Koltermann, 2007; Cheng et al., 2016), but INRs inherently lack uncertainty estimates. We address this limitation by extending the output of neural network to predict both the mean and variance of ocean temperature. Second, ocean observations are sparse and unevenly distributed (Gould et al., 2013; Boyer et al., 2016; Meyssignac et al., 2019). In our scenario, ocean observations are located sparsely, ($\sim 400$ daily profiles), and large data gaps remain if considering a $1° \times 1°$ horizontal resolution ($\sim 45,360$ grid points), as most ocean products consider (Cheng et al., 2024). We address this issue by treating each day's sparse ocean observations as a few-shot learning task. We incorporate the meta-learning framework (Hospedales et al., 2021) that learns to optimize model weights based on each day's observations. This allows MARIN to adapt to new data by one gradient step, eliminating the need for extensive retraining. By doing so, our method enables daily ocean temperature reconstructions, a significant improvement over current methods that typically offer monthly averaged estimates (Hosoda et al., 2008; Zhang et al., 2022; Cheng et al., 2024).

MARIN can reduce reconstruction errors, improve handling of sparase data and provides a flexible, continuous representation of ocean temperature fields. Experiments show that MARIN uses only 1.79% of the parameters a traditional method requires, yet reduces reconstruction error by 5.5% on unseen data when compared to in-situ observations. Notably, MARIN demonstrates significant improvements in regions with complex ocean dynamics, such as reducing RMSEs $\sim 0.2°$C in the upper 100 meters of the ocean, $\sim 0.6°$C in costal zones and high-latitude areas. These results highlight MARIN's ability to efficiently and accurately model ocean temperature fields where traditional methods struggle, making it a highly reliable choice for global ocean monitoring.

## 2 BACKGROUND AND RELATED WORK

**Objective gridded products.** Ocean gridded products are complete ocean temperature field estimates mostly produced by objective interpolation (OI) methods. So far, the most widely used products are at $1° \times 1°$ horizontal resolution and monthly temporal resolution from near-surface to about 2000 meter depths. There are amoutable ocean gridded temperature data created from various groups (Hosoda et al., 2008; von Schuckmann & Le Traon, 2011; Li et al., 2017a; Zhang et al., 2022; Cheng et al., 2024) with different data sources (Johnson et al., 2022; Cabanes et al., 2012; Boyer et al., 2018). Instead of discrete-grid values, our method produces a neural network that can represent ocean temperature at any location in the upper 2000 m of the ocean.

**Implicit representations (INRs).** Also known as coordinate-based representations, INRs offer an innovative approach to parameterizing multidimensional fields. The incorporation of Fourier features (Tancik et al., 2020) or modifications in advanced continuous function (Sitzmann et al., 2020; Ramasinghe & Lucey, 2022; Saragadam et al., 2023) further improve their representation capabilities. INRs have begun to be applied to Earth data. For example, Rußwurm et al. (2023) used spherical harmonics to encode geospatial patterns; Huang & Hoefler (2022) demonstrated over 300 times memory savings by compressing 3D weather data with MLP layers. These two papers learn representations of Earth field from densely gridded data. In

contrast, our training data here only contains raw oceanic measurements, which are sparsely and irregularly distributed.

**Usage of meta-learning in INRs.** Previous research has demonstrated the effectiveness of meta-learning for optimizing model initialization and accelerating training speed with new signals (Tancik et al., 2021; Lee et al., 2021). In this work, we adopt a similar motivation to utilize meta-learning for improving first-guess field estimated by INRs. However, we extend meta-learning usage (Finn et al., 2017; Li et al., 2017b) by employing meta-learning not only to initialize the model, but also to adapt its parameters in a single step given new in-situ observations.

**Usage of raw oceanic data in deep learning.** There are some work apply deep learning methods on raw oceanic observations. Pauthenet et al. (2022) used two-layer neural network to provide a physically consistent analysis of the upper ocean stratification in the Gulf Stream. Champenois & Sapsis (2024) used principle component analysis and temporal convolution network to reconstruct 4D temperature field in Northeast Coastal Ocean. These two works focus on regional analysis. Bagnell & DeVries (2021) uses ANNs to to interpolate and construct ocean Ocean Heat Content (OHC) to analyze warming trends. This work did analysis on a global scale, but it relies statistical approach to first construct monthly averaged fields on predefined grids. Our work learns and produces complete 4D ocean temperature field with only sparse raw oceanic observations.

## 3 METHODS

### 3.1 PROBLEM DEFINITION

The problem to produce global 4D ocean data from sparse in-situ observations is defined here. Given $N_d$ observations on day $d \in [1, D]$, $\{\mathbf{x}_i, y_i\}_{\{i=1,2,...,N_d\}}$, the objective is to learn a method $g$ that can accurately predict the ocean variables $y_q$ at any locations $\mathbf{x}_q$. Formally, we aim to solve the following optimization problem:

$$\min_\theta \sum_q \mathcal{L}\Big(y_q, g(\mathbf{x}_q | \{\mathbf{x}_i, y_i\}_{\{i=1,2,...,N_d\}}; \theta)\Big) \tag{1}$$

where, $\{\mathbf{x}_i, y_i\}_{\{i=1,2,...,N_d\}}$ is a set of $N_d$ observation locations and values pair on day $d$. $\theta$ represents parameters of model $f$ to estimate ocean field $y \approx f(\mathbf{x}; \theta)$. $\mathcal{L}$ is the loss function between predicted and true values at points $\mathbf{x}_q$. With good approximation to values anywhere on Earth, we can make complete 3D field approximation on day $d$. Iteratively optimize model $f$ day by day can thus make a time-evolving 4D temperature field.

However, in reality, we cannot measure the true values at any location. So we randomly split daily observations $\mathcal{D}_d = \{\mathbf{x}_i, y_i\}_{\{i=1,2,...,N_d\}}$ into support set $\mathcal{S}_d = \{\mathbf{x}_j, y_j\}_{\{j=1,2,...,N_\mathcal{S}\}}$ and test set $\mathcal{T}_d = \{\mathbf{x}_k, y_k\}_{\{k=1,2,...,N_\mathcal{T}\}}$. Support set here optimizes model parameters $\theta$, while test set evaluates the accuracy of the estimation made by model $f$. Therefore, Equation 1 becomes,

$$\min_\theta \sum_k \mathcal{L}\Big(y_k, g(\mathbf{x}_k | \sum_j \mathcal{L}(y_j, f(\mathbf{x}_j; \theta)))\Big) \tag{2}$$

Our problem now is to learn how to optimize model $f(\mathbf{x}; \theta)$ given support set $\mathcal{S}_d = \{\mathbf{x}_j, y_j\}_{\{j=1,2,...,N_\mathcal{S}\}}$ that makes model predict accurately on test set $\mathcal{T}_d \{\mathbf{x}_k, y_k\}_{\{k=1,2,...,N_\mathcal{T}\}}$ on day $d$.

## 3.2 FIELD REPRESENTATION WITH NEURAL NETWORK

### 3.2.1 NEURAL NETWORK STRUCTURE

For each day, ocean temperature can be seen as a scalar function of three points $y(\phi, \lambda, z)$ with latitude $\phi$, longitude $\lambda$, ocean depth $z$. MARIN proposes using neural network $f$ and learns to approximate scalar variables $y$ (e.g., temperature) from latitude-longitude-depth coordinates $\mathbf{x} = (\phi, \lambda, z) \in \Omega$ = $[-90°, 90°] \times [-180°, 180°] \times [0 \text{ m}, 2000 \text{ m}] \subset \mathbb{R}^3$. Given $N$ raw oceanic measurements with coordinate-value pairs $\{\phi_i, \lambda_i, z_i; y_i\}_{\{i=1,2,...,N\}}$, the neural network $f$ is trained with geographical locations $\{\phi_i, \lambda_i, z_i\}_{\{i=1,2,...,N\}}$ as inputs, and temperature at those locations $\{y_i\}_{\{i=1,2,...,N\}}$ as targets.

The model $f$ used in MARIN is a multilayer perceptron (MLP) with continuous activation functions. Different kinds of activation function and their implementation in MLPs is shown in Table 1.

Table 1: INRs with different activation functions. $\mathbf{W}$ and $b$ are weights and bias of a linear layer.

| Activations | Function | Layer Implementation |
|---|---|---|
| SIREN (Sitzmann et al., 2020) | $\sin(\omega_0 \mathbf{x})$ | $\sin(\omega_0(\mathbf{W}\mathbf{x} + b))$ |
| Gauss (Ramasinghe & Lucey, 2022) | $\exp(-s_0\mathbf{x}^2)$ | $\exp(-|s_0(\mathbf{W}\mathbf{x} + b)|^2)$ |
| WIRE (Saragadam et al., 2023) | $\exp(j\omega_0\mathbf{x})e(-|s_0\mathbf{x}|^2)$ | $\exp(j(\omega_0(\mathbf{W}\mathbf{x} + b)) - |s_0(\mathbf{W}\mathbf{x} + b)|^2)$ |

### 3.2.2 UNCERTAINTY ESTIMATION

A key feature of traditional objective mapping methods is their ability to provide error estimates (Cheng & Zhu, 2014b). Implicit neural representations (INRs), while powerful for approximating fields, inherently lack such uncertainty estimates since they are purely data-driven. To address this limitation, we propose a probabilistic extension to the model, enabling it to output both the mean and variance of ocean variables at each point. Specifically, we extend the output of our INR model, $f$, to predict the mean $\mu_k = f(\mathbf{x}_k; \theta)$ and variance $\sigma_k^2$ of a Gaussian distribution for the ocean variable $y$ at each spatial point $\mathbf{x}_k$:

$$p\Big(y_k|f(\mathbf{x}_k; \theta)\Big) = \mathcal{N}\Big(f(\mathbf{x}_k; \theta), \sigma_k^2\Big) \tag{3}$$

This variance $\sigma_k^2$ captures homoscedastic uncertainty, originating from data measurement errors due to instruments or locations. To train this probabilistic model, we maximize the log-likelihood of the data under the predicted distribution, leading to a maximum likelihood inference objective. The log-likelihood can be written as:

$$\log p\Big(y|f(\mathbf{x}, \theta)\Big) \propto -\frac{1}{2\sigma^2}||y - f(\mathbf{x}; \theta)||^2 - \log \sigma \tag{4}$$

For multiple observation casts, we extend this to optimize multiple outputs with uncertainties. Following the approach of Kendall et al. (2018), we use the following objective function:

$$\mu_k, \sigma_k = f(x_k; \theta), \qquad \mathcal{L}_p\Big(\mathcal{S}_d, \theta\Big) = \sum_k \frac{1}{2\sigma_k^2}||y_k - \mu_k||^2 + \sum_k \log \sigma_k \tag{5}$$

where $\mathcal{S}_d$ is the support set we defined in Section 3.1 to train our model. This uncertainty-aware loss $\mathcal{L}_p$ is the loss function we used to train our neural network described in Section 3.2.1. Prior research has demonstrated the benefits of incorporating uncertainty-aware losses for better model representations and enabling uncertainty estimation in climate models (Chen et al., 2023; Verma et al., 2024). Our approach here not only allows the model to represent the ocean field data but also provides data-driven uncertainty estimates, addressing limitation of INRs in this context.

### 3.2.3 LEARNING SINGLE STEP GRADIENT DESCENT

The accuracy of the estimated ocean field $f(\mathbf{x}; \theta)$ depends on the ability of the model to adapt to new in-situ observations. However, daily raw oceanic data ($\sim 400$ profiles per day) is insufficient to train INRs effectively from scratch, at the risk of overfitting. To overcome this problem, we draw inspiration from few-shot learning problems (Ravi & Larochelle, 2016; Parnami & Lee, 2022), where the neural network learns to represent ocean temperature accurately from a few observed data. We, therefore, define the procedure to reconstruct the complete ocean field without retraining:

1. Given in-situ observations on each day $d$, randomly split all observations $\mathcal{D}_d$ into support set $\mathcal{S}_d$ and test set $\mathcal{T}_d$.

2. Predict temperature given coordinates from support set $\mathcal{S}_d$: $\{f(\mathbf{x}_S; \theta) | \{\mathbf{x}_S, y_S\} \in \mathcal{S}_d\}$.

3. Use Equation 5 to compute the uncertainty-aware loss $\mathcal{L}_p$ given model estimates $f(\mathbf{x}_S; \theta)$ and observed values $y_S$ in support set.

4. Perform one-step gradient descent: $\theta^* = \theta - \alpha \circ \nabla_\theta \mathcal{L}_p(\mathcal{S}_d, \theta)$.

5. Make complete ocean field approximation with updated parameters $\theta^*$: $f(\mathbf{x}; \theta^*)$.

6. Evaluate field approximation with test set $\mathcal{T}_d$ by computing the mean squared error (MSE) between approximation and true values: $\mathcal{L}_2\Big(y_T, f(\mathbf{x}_T; \theta^*)\Big) = \Big(y_T - f(\mathbf{x}_T; \theta^*)\Big)^2$, $\{\mathbf{x}_S, y_S\} \in \mathcal{T}_d$.

Here, $\nabla_\theta \mathcal{L}_p$ is the gradient of $\mathcal{L}_p(\theta)$ and $\circ$ is the element-wise dot product. $\alpha$ is a trainable vector of the same size as $\theta$ that decides the step size of the update. We now can modify our optimization problem previously written in Equation 2,

$$\min_{\theta, \alpha} \sum_k \mathcal{L}_2\Big(y_k, f(\mathbf{x}_k; \theta - \alpha \circ \nabla_\theta \mathcal{L}_p(\mathcal{S}_d, \theta))\Big) \tag{6}$$

With $N$ days' unevenly distributed daily profiles, we thus need to learn how to optimize model parameters $\theta$ with gradient step size $\alpha$. This "learn how to optimize" problem aligns with the goal of meta-learning (Hospedales et al., 2021). Our meta-learning framework has two optimization loops, which follows Finn et al. (2017) and Li et al. (2017b). The inner-loop samples and optimizes daily profiles on $N$ days, while the outer-loop. Other implementation details related to our meat-learner are presented in Appendix B.2.

We summarize the algorithm to train our model in Algorithm 1. This dynamic adjustment corresponds to OI's bias correction procedure with error covariance function, but with the added advantage of learning across daily profiles for better generalization. Algorithm 2 summarizes the procedure to produce ocean data given in-situ observations. By incorporating meta-learning with a task structure that treats daily ocean fields as separate tasks, we enable fast adaptation with a single step optimization given new daily observations.

---

**Algorithm 1** Meta-Learning for Projection Function

---

**Require:** In-situ observations $\{\mathbf{x}_i, y_i\}_{i=1}^N$, meta-optimizer learning rate $\beta$
**Require:** Trainable parameters: projection function $\theta$ and same size learnable step size $\alpha$.
1: **while** not done **do**
2:     **for** each day $d$ **do**
3:         Split daily observations $\{\mathbf{x}_i, y_i\}_{i=1}^N$ into support set $\mathcal{S}_d$ and target set $\mathcal{T}_d$
4:         **Inner loop**: update $\theta$ on support set $\mathcal{S}_d$: $\theta'_d = \theta - \alpha \circ \nabla_\theta \mathcal{L}_p(\mathcal{S}_d, \theta)$
5:     **end for**
6:     **Outer loop**: Compute meta-loss on target set $\mathcal{T}_d$: $\mathcal{L}_{\text{meta}} = \sum_d \mathcal{L}_p(\mathcal{T}_d, \theta'_d)$
7:     Update $(\theta, \alpha) \leftarrow \theta - \beta \nabla_{(\theta, \alpha)} \mathcal{L}_{\text{meta}}$
8: **end while**

---

---

**Algorithm 2** Real-Time Field Generation Using Trained Model

---

**Require:** Query coordinates $\mathbf{x}_q$, in-situ observations $\{(\mathbf{x}_i, y_i)\}_{i=1}^{N}$, trained parameters $\theta^*$ and $\alpha^*$
 1: **Step 1: Adapt to Observations**
 2: For each observation $\mathbf{x}_i$, compute prediction: $\mu_i, \sigma_i{}^2 = f(\mathbf{x}_i; \theta^*)$
 3: Update parameters $\theta^*$ using: $\theta^* \leftarrow \theta^* - \alpha^* \circ \nabla_\theta \mathcal{L}_p(\theta^*)$
 4: **Step 2: Estimate Field at Query Points**
 5: **for** each query point $\mathbf{x}_q$ **do**
 6:     Estimate field: $\mu_q, \sigma_q{}^2 = f(\mathbf{x}_q; \theta^*)$
 7:     Generate field: $y_q \sim \mathcal{N}(\mu_q, \sigma_q{}^2)$
 8: **end for**
 9: Return estimated field $y_q$ at all query points

---

## 4 EXPERIMENTS

SIREN (Sitzmann et al., 2020) with 2 hidden layers and 128 hidden neurons is the primary architecture we used to approximate temperature field. Given that conventional mapping methods often use 12-month climatologies for their initial guess fields (Smith & Murphy, 2007; Levitus et al., 2012), we train a separate model for each month (twelve models in total). We downloaded raw oceanic measurements from the World Ocean Database (WOD) (Boyer et al., 2018) and trained our model on data collected over 15 years, from 2006 to 2020. This timeframe was chosen because the expansion of the Argo array has enabled near-global sampling of the upper 2000 meters of ice-free oceans since 2005, providing a rich dataset with minimal bias since 2006 (Roemmich et al., 2019). Additionally, research indicates that using a shorter 15-year period is preferable (Cheng & Zhu, 2015). Longer datasets can introduce inconsistencies in spatial baselines across locations, which could violate the spatial structure of the temperature field (Cheng & Zhu, 2015; Li et al., 2022). Further implementation details and hyperparameters are provided in Appendix B.

### 4.1 DATA

**Observational data sources.** Among global datasets, the most comprehensive and widely used is the World Ocean Database (WOD) produced by the National Oceanographic Data Centre (NODC) Ocean Climate Laboratory (OCL) (Boyer et al., 2018). This work collected raw observational data from WOD between 2000 and 2022. Data from all instrument types are used, including Argo profiling floats (PFL) (Argo, 2000), eXpendable bathythermographs (XBTs) (Goni et al., 2019), conductivity-temperature-depth (CTD), mechanical bathythermographs (MBTs), moored buoy (MRB), drifting buoy (DRB), animal-borne ocean sensors (APB) (McMahon et al., 2021) and gliders (GLD) (Boyer et al., 2018). In this paper, data from WOD is used to reconstruct complete daily ocean temperature fields and evaluate accuracy of reconstruction against traditional methods.

**Satellite observations.** Additionally, we use Global Ocean OSTIA Sea Surface Temperature and Sea Ice Analysis (OSTIA) (Good et al., 2020), which combines in-situ and satellite data from infrared and microwave radiometers to produce daily sea surface temperature (SST) maps since 2007 at a higher horizontal resolution $0.05° \times 0.05°$. In this study, OSTIA serves as the ground truth for evaluating our data products and comparing them against other datasets.

**Competing methods.** In this study, we validate our results using three gridded observational datasets: BOA-Argo (Li et al., 2017a), GDCSM-Argo (Zhang et al., 2022), and IAPv4 (Cheng et al., 2024). Details of these datasets are in Appendix A, Table 5. These datasets were selected for their relevance and contemporary methodologies, as we avoid comparing with methods published before 2015 to ensure a fair evaluation against current objective mapping techniques. We also exclude some open-source data due to un-

availability, insufficient vertical levels, or incomplete temporal coverage between 2005 and 2022, ensuring our comparisons are robust and meaningful.

### 4.2 Evaluation model reconstruction accuracy

#### 4.2.1 Subsample tests

Since we are unable to measure ocean temperatures anywhere on Earth at any time, we employ a subsample testing approach to systematically assess our model's reconstruction accuracy and compare its performance against the three open-sourced ocean temperature products mentioned above. In a single subsample test, observational data from a certain day is split into a support set and a test set using a 80/20 ratio. The support set, comprising 80% of the available observations, is used to optimize model parameters and reconstruct a complete ocean temperature following our proposed algorithm 2. The test set, consisting of the remaining 20%, is reserved for evaluation. Specifically, the model's estimated field values at test set locations are compared against the true observed values, and we compute the root mean squared error (RMSE) to quantify reconstruction accuracy,

$$\text{RMSE} = \sqrt{\frac{1}{N_T} \sum_{T=1}^{N_T} \left( y_T - f(\mathbf{x}_T, \theta) \right)^2}, \qquad \{\mathbf{x}_T, y_T\}_{T=1,2,\dots,N_T} \in \mathcal{T} \tag{7}$$

where $N_T$ is the total number of samples in the test set $\mathcal{T}$. $y_T$ are the true values at the test locations and $f(\mathbf{x}, \theta)$ are the model predicted values. We repeat this subsample test 100 times per day over the entire evaluation period (January, 1, 2000 - December 31, 2022), with random sampling of support and test sets for each repetition. In total, this results in 837,200 subsample tests, providing a robust evaluation of the MARIN's performance across time and varying locations. A more detailed analysis is done by splitting the evaluation period (2000-2022) into three evaluation intervals:

- **2000-2005**: Data is limited and is not used to train our model.
- **2006-2020**: This period corresponds to the model's training window, during which the model was optimized using observational data.
- **2021-2022**: This includes observations recently but unused for training model.

The division into these three intervals allows us to analyze MARIN's performance in different scenarios. The 2000-2005 period tests the model's robustness during time of limited observations. The 2006-2020 validates the model's reconstruction capabilities. The 2021-2022 period offers a challenging assessment of the model's ability to generalize to new data, reflecting real-world applicability.

#### 4.2.2 Results and analysis

We evaluated the reconstruction accuracy of MARIN using subsample tests across different periods and ocean locations. We do not evaluate performances of BOA-Argo and GDCSM-Argo in 2000-2005 period because their temporal coverage is after 2004. Table 2 reveals MARIN's supreme performance compared with existing methods. MARIN achieved the lowest RMSE ($0.858°C$) during the 2006-2020 training period, outperforming BOA-Argo ($1.070°C$), GDCSM-Argo ($1.217°C$) and IAPv4 ($1.030°C$). MARIN also has the best performance across the two unseen periods (2000-2005, 2021-2022), demonstrating its strong generalization ability. When considering monthly RMSEs, MARIN have notable improvements in general, receiving the lowest RMSE in almost all months in both period 2006-2020 and period 2021-2022 (see Appendix C Table 7 and Table 8). These results confirm that MARIN not only excels in reconstructing ocean temperatures from sparse data but also generalizes well to unseen data, making it a highly reliable choice to support global ocean monitoring.

Table 2: Monthly $\mathrm{RMSE}(\downarrow)$ of data products based on observation measurements between 2000 and 2022.

| Time coverage | MARIN(Ours) | BOA-Argo | GDCSM-Argo | IAPv4 |
|---|---|---|---|---|
| 2006-2020 | 0.858 | 1.070 | 1.217 | 1.030 |
| 2000-2005 | 1.044 | - | - | 1.055 |
| 2021-2021 | 1.001 | 1.059 | 1.202 | 1.077 |

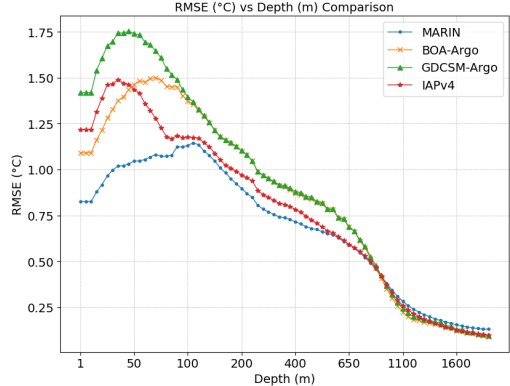

Further analysis is made to determine locations vertically and horizontally MARIN makes significant improvements compared with traditional OI methods.

Reconstruction errors at different ocean depths are shown in Figure 1. Our method (MARIN) consistently achieves the lowest RMSEs in the upper 500 m of the ocean, with significant improvements in the upper 100 m. Previous studies have shown that the upper 100 m of ocean presents highly nonlinear temperature variations, driven by sunlight and geographic factors (Barrier et al., 2023; Lin et al., 2023). Variations here make it particularly challenging to estimate. The higher RMSE values in this zone also validate the complexity of the temperature dynamics. Remarkable improvement of approximation here by MARIN indicates that our neural-network-based approximation is effective to address the difficulties of estimating regions with complex temperature patterns.

Figure 1: Vertical distributions of temperature RMSE (°C) as a function of ocean depths. MARIN (our method) here exhibits the lowest RMSEs at most depths, particularly in the upper 500 m of the ocean.

Horizontal RMSE distributions of MARIN and three other ocean products are shown in Figure 2. In the top row, we observe that for all methods, the distributions of temperature RMSEs follow a similar pattern, with higher errors in coastal regions and areas with complex ocean dynamics, such as the Gulf Stream (Hurlburt et al., 2011; Mensa et al., 2013) and the Southern Ocean. The bottom row here visualizes RMSE (°C) differences between MARIN and BOA-Argo, GDCSM-Argo, and IAPv4. The red shading here indicates regions where MARIN outperforms competing methods. MARIN maintains lower RMSE values in most regions. especially in coastal areas and high-latitude zones, reducing errors over $0.7°C$, suggesting it handles these dynamics areas better than competing methods. A similar result is found when we evaluate RMSEs based on gridded satellite observations, where MARIN has significantly lower RMSEs in high-latitude and coastal regions (see Figure 5 in Appendix).

### 4.3 EFFICIENCY OF MARIN

The traditional OI method to represent ocean fields by discrete grid cells requires heavy memory storage. For example, BOA-Argo has a horizontal $1° \times 1°$ and 58 vertical levels, which needs $58 \times 180 \times 360 = 3.758M$ grid values to store for each time step. In contrast, our method only contains 0.068M parameters and has smaller construction errors as shown in previous experiments. Table 3 shows the required parameters of MARIN and each global ocean product we compare against. We can see that not only MARIN is resolution-free both horizontally and vertically, but it also reduces parameters over 50 times, making our method efficient to use.

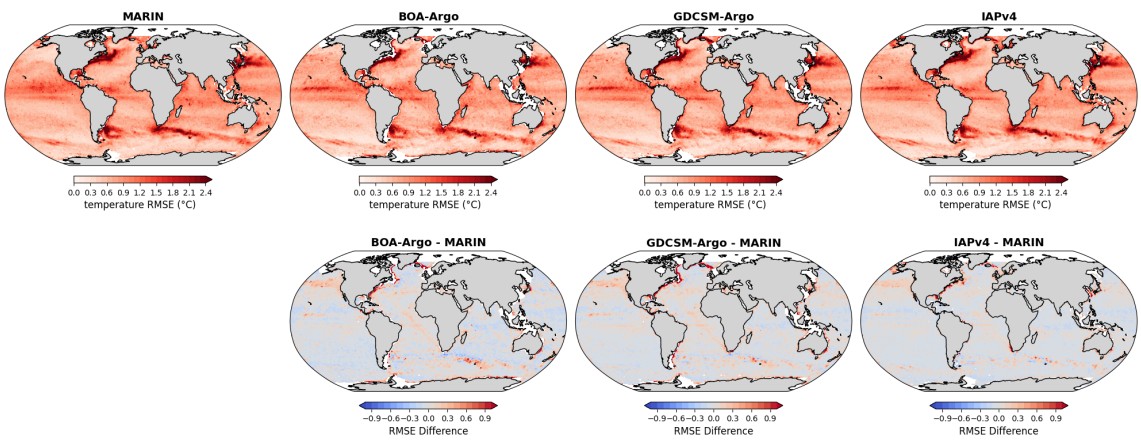

Figure 2: Horizontal distribution of temperature RMSE (°C), across MARIN (ours), BOA-Argo, GDCSM-Argo, IAPv4 (top row). RMSE (°C) differences between MARIN and the three competing methods (bottom row). MARIN achieves lower RMSEs across most ocean regions, especially in complex, high-variability zones like high-latitude areas and coastal regions (red shaded areas in bottom row).

Table 3: Settings and number of parameters of different methods.

| Methods | Resolution | Levels | Params (M) |
|---|---|---|---|
| BOA-Argo | $1° \times 1°$ | 58 | 3.758 |
| GDCSM-Argo | $1° \times 1°$ | 58 | 3.758 |
| IAPv4 | $1° \times 1°$ | 79 | 5.119 |
| MARIN (Ours) | unlimited | unlimited | 0.068 |

## 5 ABLATION STUDIES

**Effectiveness of Meta-Learning.** To check the effectiveness of our meta-learning framework proposed in Section 3.2.3, we make additional experiments to train our INRs model on daily observations from scratch. On each day we take 80% of data for training and reserve 20% to compute test RMSEs. Following our experiments, we repeat these random sampling and training experiments 100 times per day over the period (2006-2022). Results here in Table 4 shows that meta-learning is important, because our daily observations are limited. Incorporate meta-learning into MARIN will greatly improve our models reconstruction accuracy and generalisation to unseen data period (i.e., 2021-2022). Training from scratch day by day will make our method worse if compared against traditional methods.

Table 4: Comparison on averaged test RMSE (°C) between meta-learning and training data from scratch.

| Period | train from scratch | meta-learning one step gradient |
|---|---|---|
| 2006-2020 | 1.539 | 0.858 |
| 2021-2022 | 1.555 | 1.001 |

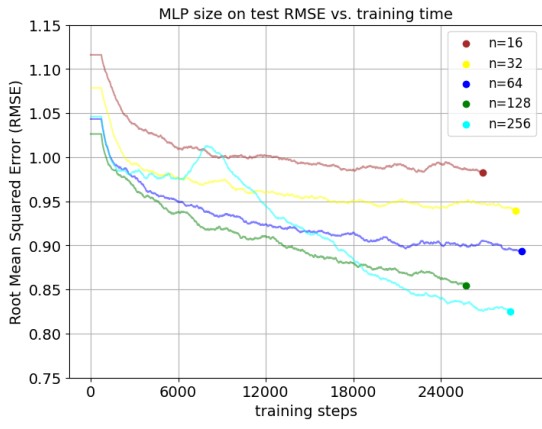 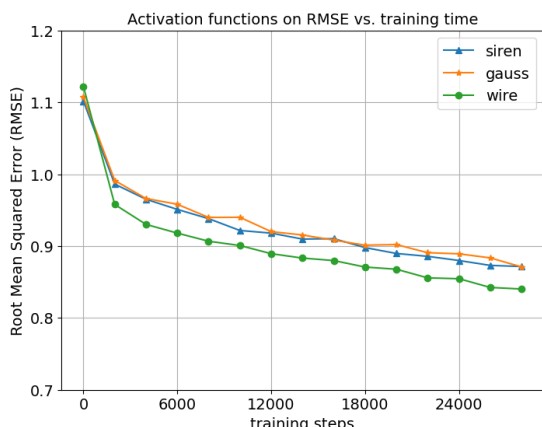

Figure 3: MARIN 's test temperature RMSE (°C) as a function of training steps with different MLP size.

Figure 4: MARIN 's test temperature RMSE (°C) with different activation functions.

**Size of MLP layers.** Figure illustrates the relationship between the size of the MLP layers and the test error over training epochs. As shown in Figure 3, increasing the number of hidden dimensions generally reduces reconstruction errors. However, we observed instability in training when using larger MLP sizes, such as a significant and irregular increase in errors after 6,000 training steps for an MLP size of 256. Additionally, larger models demand greater memory resources. Consequently, we conducted our experiments with MaIIN using an MLP size of 128, striking a balance between performance and stability.

**Different activation functions.** In addition to the SIREN network, we explored two other continuous activation functions: Gaussian (Ramasinghe & Lucey, 2022) and Gabor wavelet (WIRE) (Saragadam et al., 2023). All networks trained consist of 2 layers with a hidden size of 128. Figure 4 shows the experiments we made on three different INRs, plotting the RMSE on test set against training steps. All methods have good performance in general. WIRE network exhibited a faster learning rate and achieved the lowest RMSE compared to both the SIREN and Gaussian networks. These findings are consistent with WIRE's performance as reported in state-of-the-art signal representation studies (Saragadam et al., 2023).

# 6 CONCLUSION AND FUTURE WORK

In this paper, we introduced a novel method MARIN, to reconstruct dynamic 4D ocean fields using implicit neural representations (INRs) enhanced with uncertainty estimates and meta-learning. Taking advantages of the continuous, nonlinear modeling capabilities of neural networks, our method demonstrated superior performance over traditional interpolation techniques, offering daily, resolution-free ocean temperature reconstructions. As more ocean observation data becomes available, our approach is expected to yield even more accurate temperature reconstructions. We hope the contributions of this paper will inspire further advancements in AI-driven ocean field estimation. While our method shows substantial improvements, several promising directions remain for future exploration. Expanding the model to include additional ocean variables, such as salinity and currents, would enhance the comprehensiveness of the reconstructions. Additionally, improving the handling of extreme weather events and seasonal anomalies could make the model more resilient to abrupt environmental changes. Tackling these challenges will contribute to more complete weather and climate analyses and benefit the society as a whole.

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

## A DATA

### A.1 DATA PRODUCTS INCLUDED FOR COMPARISON

Information of three ocean-gridded datasets we compared against in this paper is listed in Table 5. All these datasets are at global scale, created independently and updated monthly, at $1° \times 1°$ horizontal resolution and monthly temporal resolution. Sources of data to make their products are different, where BOA-Argo and GDCSM-Argo used Argo profiling floats and IAPv4 used data from WOD and extend analysis to 6500 meters below sea surface.

Table 5: Details of gridded data products published after 2015 for comparison.

| Name | BOA-Argo | GDCSM-Argo | IAPv4 |
|---|---|---|---|
| Area | $180°W - 180°E,$ $90°S - 90°N$ | $180°W - 180°E,$ $80°S - 80°N$ | $180°W - 180°E,$ $90°S - 90°N$ |
| Resolution | $1° \times 1°$ | $1° \times 1°$ | $1° \times 1°$ |
| Levels | 0-2000 m, 58 levels | 0-2000 m, 58 levels | 0-6500 m, 119 levels |
| Temporal coverage | since 2004 | since 2004 | since 1940 |
| Temporal resolution | monthly | monthly | monthly |
| | Shanghai Ocean University | China Second Institute of Oceanography (CSIO) | Institute of Atmospheric Physics (IAP) |
| Reference | Li et al. (2017a) | Zhang et al. (2022) | Cheng et al. (2024) |

### A.2 DATA PREPROCESSING

Data preprocessing here aims to eliminate duplicate observations, correct measurement bias, filter out bad quality data. After data preprocessing, observational profiles are grouped by date and we therefore have a high-quality historical daily observational data between 2000 and 2022.

#### A.2.1 DUPLICATE CHECK

The duplicate check looks for pairs within $0.1°$ longitude and latitude and 1 day when their instrument types are different (e.g. PFL and XBT). This might happen when two types of measurement profiles occasionally locate extremely close.

### A.2.2 DATA QUALITY CONTROL

The quality control (QC) procedure is important for precise temperature reconstruction. This work's QC procedure follows the CAS-Ocean Data Center (CODC) Quality Control system (CODC-QC) (Tan et al., 2023), which has already been used to create high-quality ocean gridded data (Zhang et al., 2024; Cheng et al., 2024). Table 6 lists the 14 QC checks proposed by Tan et al. (2023) to filter bad data. In the CODC-QC, the definition of the QC flag is dichotomic for each check at each observed depth, with 0 denoting the acceptable (good) value and 1 indicating the rejected (bad) value. A final quality flag is also provided based on all distinct quality checks. This work only use data with final quality flag equals 0 for training and evaluation.

Table 6: Details of each QC check from CODC-QC.

| Order | Name of checks | Descriptions |
|-------|----------------|--------------|
| (1) | Basic information check | Check whether date, time, location are in acceptable ranges. |
| (2) | Sample level order check | Check whether profile sample depths increase monotonically. |
| (3) | Instrument maximum depth check | Check whether sample depth exceeds instrument type maximum depth. |
| (4) | Local bottom depth check | Check whether sample depth exceeds local bottom depth. |
| (5) | Global range check | Check whether temperature falls outside global limits. |
| (6) | Sea-water freezing point check | Check whether temperature is lower than the freezing point temperature. |
| (7) | Constant value check | Check temperature profile for unrealistic thermostats. |
| (8) | Local climatological range check | Check whether temperature falls outside the local climatological limits. |
| (9) | Spike check | Check whether temperature observation is a spike value. |
| (10) | Density inversion check | Check whether the density increases with increasing depth. |
| (11) | Multiple extrema check | Check whether the profile exhibits an unrealistic number of temperature extrema. |
| (12) | Global vertical gradient check | Check whether vertical temperature gradient falls outside the global limits. |
| (13) | Local gradient climatological range check | Check whether vertical temperature gradient falls outside the local climatological limits. |
| (14) | Instrument specific check | Only checks in XBT profiles related to wire stretch, leakage, hit bottom, wire break, etc. |

## B IMPLEMENTATION DETAILS

### B.1 DATA NORMALIZATION

Input coordinates, including latitude, longitude, and ocean depth are each normalized to 0-1 range by min-max normalization. For temperature values, following conventional settings (Cheng et al., 2017; 2024), compute temperature anomalies by removing long-term monthly climatological means. We notice that the quality of the climatology field is crucial for global ocean reconstruction (Cheng & Zhu, 2014b; Boyer et al., 2016). In this paper, we use the 30-year (1991-2020) $1° \times 1°$ monthly means from World Ocean Atlas 2023 (WOA23) (Boyer et al., 2018) as our climatological field. This is because WOA23 create monthly means based on data from WOD, which is the same source as our training data and previous works have use world ocean atlas as ground truth values for ocean climatology analysis (Rohling et al., 2015; Meyssignac et al., 2019; Osman et al., 2021).

### B.2 META-LEARNING SETTINGS

The task here for meta-learning is to learn how to reconstruct a complete ocean field from ocean in-situ observations each day. Each task here contains daily observations on a single day, and is split into support/test set in a 80/20 ratio. However, the number of tasks we have is efficient to train a meta-learner. For example, the 15-year training period (2006-2020) has $31 \times 15 = 465$ tasks if we want to train a monthly model for January. This is a heavy burden for meta-learning since we need to wait for 465 tasks and perform a single meta-optimizing step on model parameters $\theta$ and gradient step $\alpha$. Therefore, for each step in the outer loop, we sample data from one day in each month and create a batch of data containing in-situ observations from 15 days (i.e., 15 tasks). The reduction from 465 tasks to 15 further accelerates our training process, making our method easy to train.

### B.3 HYPERPARAMETERS

We used Cosine-Annealing Learning Rate scheduler, with maximum learning rate 1.e-3 and minimum learning rate 1.e-6. We trained the outer-loop of our meta-learning for 2000 epochs.

### B.4 SOFTWARE AND HARDWARE

The model and meta-learning pipeline is implemented in PyTorch (Paszke et al., 2019). The whole model training and inference is conducted on a single 40GB NVIDIA A100 device.

## C SUPPLEMENTARY RESULTS

This section provides additional results to support our evaluation.

### C.1 EVALUATION ON DATA FROM WOD

In addition to the averaged RMSE of all months, we separately tested RMSEs of twelve different months at different evaluation period. Table 7 and Table 8 provides the monthly RMSE of different interpolation methods between 2006-2020 and 2021-2022, respectively. We can see that for almost all months (12 for period 2006-2020, 9 for 2021-2022) MARIN has the lowest RMSEs compared with three competing datasets. This results demonstrates that our method has stable performances in any season.

Table 7: Monthly RMSE($\downarrow$) of data products based on WOD observations from 2006 to 2020.

| Time coverage | Month | MARIN (Ours) | BOA-Argo Li et al. (2017a) | GDCSM-Argo Zhang et al. (2022) | IAPv4 Cheng et al. (2024) |
|---|---|---|---|---|---|
| | Jan | **0.855** | 1.001 | 1.093 | 0.999 |
| | Feb | **0.829** | 1.067 | 1.242 | 1.003 |
| | Mar | **0.756** | 0.954 | 1.089 | 0.927 |
| | Apr | **0.792** | 0.994 | 1.127 | 0.910 |
| | May | **0.918** | 1.096 | 1.185 | 1.130 |
| | Jun | **0.873** | 1.053 | 1.275 | 1.053 |
| 2006-2020 | Jul | **0.902** | 1.097 | 1.275 | 1.168 |
| | Aug | **0.904** | 1.127 | 1.395 | 1.140 |
| | Sep | **0.851** | 1.099 | 1.125 | 0.987 |
| | Oct | **0.907** | 1.134 | 1.263 | 1.069 |
| | Nov | **0.866** | 1.184 | 1.285 | 0.993 |
| | Dec | **0.843** | 1.027 | 1.107 | 0.945 |
| | **Overall.** | **0.858** | 1.072 | 1.218 | 1.026 |

Table 8: Monthly RMSE($\downarrow$) of data products based on WOD observations from 2021 to 2022.

| Time coverage | Month | MARIN (Ours) | BOA-Argo Li et al. (2017a) | GDCSM-Argo Zhang et al. (2022) | IAPv4 Cheng et al. (2024) |
|---|---|---|---|---|---|
| | Jan | **1.028** | 1.033 | 1.199 | 1.045 |
| | Feb | **0.869** | 1.014 | 0.943 | 0.999 |
| | Mar | **0.950** | 0.977 | 1.159 | 1.098 |
| | Apr | **0.925** | 0.985 | 1.142 | 0.981 |
| | May | 0.942 | 0.942 | 0.974 | **0.936** |
| | Jun | **0.966** | 0.983 | 1.217 | 1.029 |
| 2006-2020 | Jul | 1.039 | **1.012** | 1.524 | 1.197 |
| | Aug | **1.221** | 1.295 | 1.410 | 1.331 |
| | Sep | **0.990** | 1.094 | 1.245 | 1.079 |
| | Oct | **1.014** | 1.196 | 1.154 | 1.051 |
| | Nov | **1.035** | 1.229 | 1.320 | 1.129 |
| | Dec | 0.953 | 1.006 | 1.006 | **0.942** |
| | **Overall.** | **1.001** | 1.056 | 1.198 | 1.077 |

## C.2 EVALUATION BASED ON SATELLITE OBSERVATIONS

We also assessed the accuracy of our method in reconstructing sea surface temperature (SST) using the OSTIA dataset (Good et al., 2020), which is often employed as truth values for observational temperature data at the sea surface (Ferreira et al., 2024; Barbosa Aguiar et al., 2024). The overall reconstruction errors are presented in Table 9. MARIN is the second-best methods, slightly worse than GDCSM-Argo. We attribute this to two reasons: (1) no efficient surface observations for learning; and (2) data distribution mismatch between instrument measurements and satellite observations (Zhang et al., 2022).

The horizontal distributions of RMSE are presented in Figure 5. Consistent with our findings from the WOD data, our method demonstrates reduced errors at high latitudes (red shaded regions), highlighting the improvements in representations where observations are sparse and the topography is complex. However, we

Table 9: Monthly RMSE(↓) of data products based on satellite SST between 2007 and 2022.

| Time coverage | MARIN | BOA-Argo | GDCSM-Argo | IAPv4 |
|---|---|---|---|---|
| 2007-2022 | 0.878 | 0.917 | 0.863 | 0.881 |

observed increased errors in equatorial regions. As noted by Zhang et al. (2022), there is a current deficiency in measurements from profiling floats at the sea surface. Relying on data from other instruments introduces errors due to mismatched biases and varying quality. The challenge of balancing data usage with appropriate interpolation methods remains an open problem for accurately estimating ocean surface temperatures.

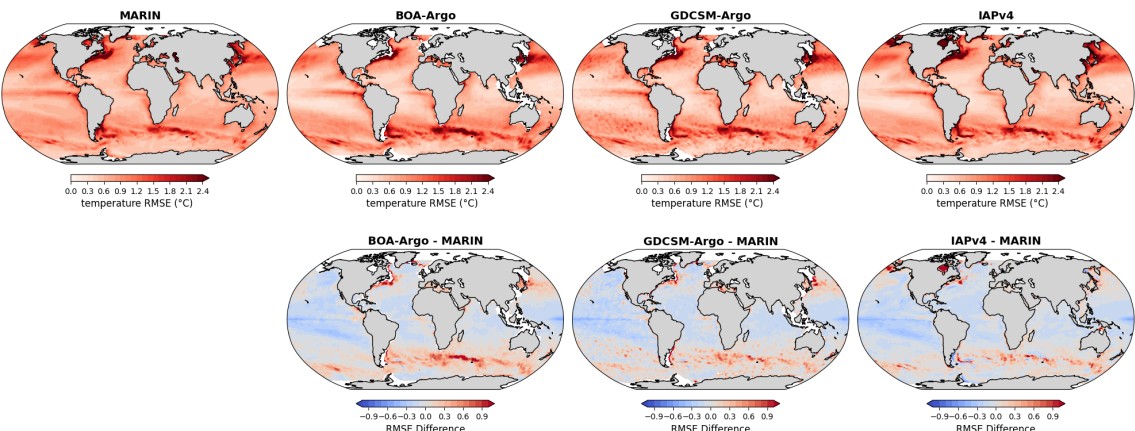

Figure 5: Horizontal distributions of RMSE (°C) based on satellite gridded SST (OSTIA) (top row) and comparison against three other data products (bottom row).

