# OpenReview forum: "Transforming Ocean Analysis: Learning 4D ocean field from in-situ observations via uncertainty-aware implicit representations"
_ICLR.cc/2025/Conference — ICLR 2025 Conference Withdrawn Submission_

### Official Review · Reviewer_5FcN · 2024-10-28

**Soundness:** 2
**Presentation:** 2
**Contribution:** 2
**Rating:** 3
**Confidence:** 4

**Summary:**

This paper presents MARIN, a model for reconstructing 4D ocean temperature fields from sparse in-situ data by combining implicit neural representations with meta-learning and uncertainty estimation. MARIN treats ocean temperatures as a continuous function, extending INRs to predict both mean and variance for added uncertainty assessment. Although it adapts these techniques for real-time ocean analysis, the model’s design largely relies on existing INR and meta-learning frameworks, without substantial advancements in core architecture. Limited novelty lies in this application.

**Strengths:**

- Unlike objective interpolation methods that rely on predefined grids, MARIN models the ocean temperature field as a continuous function
- To account for data noise and uncertainty in ocean observations, MARIN extends the INR to predict both the mean and variance of temperature.

**Weaknesses:**

- MARIN uses implicit neural representations without significant architectural advancements. Improved geospatial encoding or spherical coordinate modeling [1,2,3], rather than reliance on standard baselines not optimized for sparse data, could enhance its performance.
- MARIN captures only aleatoric uncertainty, omitting epistemic uncertainty, which is critical for complex, sparse data. Including both types would provide a more comprehensive uncertainty analysis.

1. Kim, H., Jang, Y., Lee, J. and Ahn, S., Hybrid Neural Representations for Spherical Data. In Forty-first International Conference on Machine Learning.
2. Mai, G., Lao, N., He, Y., Song, J. and Ermon, S., 2023, July. Csp: Self-supervised contrastive spatial pre-training for geospatial-visual representations. In International Conference on Machine Learning (pp. 23498-23515). PMLR.
3. Bonev, B., Kurth, T., Hundt, C., Pathak, J., Baust, M., Kashinath, K. and Anandkumar, A., 2023, July. Spherical fourier neural operators: Learning stable dynamics on the sphere. In International conference on machine learning (pp. 2806-2823). PMLR.

**Questions:**

- MARIN’s design overlooks temporal dynamics, which seems surprising for modeling something as continuously evolving as ocean temperatures. Including these effects could enhance its performance, and comparing it with models [1] that do account for temporal dynamics would clarify how much this omission impacts results.
 - Using negative log-likelihood for uncertainty can sometimes miscalibrate variance, especially with extreme noise. I suggest testing MARIN across varying noise levels to show how reliable its uncertainty estimates truly are.
- Current experimental settings seem a bit confusing to me. Could you report whether MARIN’s uncertainty estimates hold up consistently across different data splits, particularly with random splits?
- I wonder if a simpler adaptation approach, like daily fine-tuning without meta-learning, could yield similar results. An ablation study comparing this with the current meta-learning approach would help determine if meta-learning is truly essential for MARIN’s reported performance.

1. Williams, J.P., Zahn, O. and Kutz, J.N., 2024. Sensing with shallow recurrent decoder networks. Proceedings of the Royal Society A, 480(2298), p.20240054.

---

### Official Review · Reviewer_Z9NN · 2024-10-31

**Soundness:** 1
**Presentation:** 2
**Contribution:** 2
**Rating:** 3
**Confidence:** 3

**Summary:**

This paper deals with the data assimilation problem in oceanography, where the aim is to reconstruct the ocean field from sparse observations. The authors propose a deep learning model for this problem based on implicit neural representations, and cast the data assimilation problem as a few-shot meta-learning training objective. The learning model is evaluated on real-world a large time period.

**Strengths:**

Data assimilation is an important problem, especially for the ocean where the number of observations is typically very low. This problem is well motivated in the paper, with its implications tied to climate change.

The meta-learning framework is clearly introduced. Neural fields are a promising learning architecture for the data assimilation problem, as they allow to model the geophysical state with a continuous parametric function of space.

Experiments are conducted on real-world data and the proposed method demonstrates competitive results in terms of reconstruction error, at different ocean depths and for different time periods.

The computational cost of the method is studied. Since implicit neural representations parametrize the unknown state as a parametric function of space, they offer a resolution-agnostic representation of the physical field, leading to considerable computational savings, as pointed out in Section 4.3. This is a significant advantage of using neural fields over other grid-based neural representations.

Ablation studies support the relevance of using meta-learning, and study the effect of the model architecture on performance.

**Weaknesses:**

My main concern is that this paper does not properly present the reconstruction problem under study, with no mention of "data assimilation" or "optimal interpolation". I feel that the authors may be misusing the term "objective interpolation", in place of “optimal interpolation”.The paper also lacks reference to the state-of-the art techniques in this field, including 3D-Var, 4D-Var and Kalman filters. In the "Competing methods" paragraph, the authors mention the datasets but not the interpolation techniques used, which is ultimately what they compare against. This lack of precision makes it difficult to assess the submission's contribution.
I suggest that the authors explicitly compare their method with traditional state-of-the-art data assimilation methods and recent deep-learning-based approaches [1, 2, 3].

If I understood correctly, the proposed method consists in applying MAML to ocean data,treating different days as tasks to predict, and with the last layer predicting a Gaussian distribution rather than a scalar prediction.. Given that the primary novelty lies in predicting a distribution rather than simply a field, I believe more insight should be provided on the method’s uncertainty quantification capabilities, which are mentioned but not well illustrated. How does the proposed method lend itself to building confidence intervals, or to sampling from a posterior predictive distribution? Can uncertainty quantification be illustrated on a toy example, and compared with the existing methods?

At test time, only a single gradient step is performed to adapt the network, as mentioned in Algorithm 2. Why not more? In meta-learning, while the number of inner gradient steps is often chosen to be 1 at training time for computational reasons, it can be larger at prediction time. Further, the adaptation procedure of Algorithm 2 where the ocean state is reconstructed by tuning the model parameters should be compared with that of traditional variational data assimilation, where the state is reconstructed by maximizing the posterior likelihood.

Is it relevant to train 12 separate models—one per month, as indicated on line 251? Why not train a single global model using all available data? Additionally, it would be useful to discuss how the temporal dimension of the signal is handled. If I understand correctly, the proposed neural fields are functions only of space, with time treated as multiple independent tasks across days. What if there are observations at different times within the same day? Could the model incorporate time correlations by introducing a time variable or embedding in the architecture?

**Questions:**

The meaning of "Projection function" in Algorithm 1 is unclear.

There are a number of clarity issues and typos in the paper.

### References

[1] Fablet, R., Huynh Viet, P., Lguensat, R., Horrein, P. H., & Chapron, B. (2018). Spatio-temporal interpolation of cloudy SST fields using conditional analog data assimilation. Remote Sensing, 10(2), 310.

[2] Blanke, M., Fablet, R., & Lelarge, M. (2024). Neural Incremental Data Assimilation. arXiv preprint arXiv:2406.15076.

[3] Rozet, F., & Louppe, G. (2023). Score-based data assimilation. Advances in Neural Information Processing Systems, 36, 40521-40541.

---

### Official Review · Reviewer_Kzma · 2024-11-02

**Soundness:** 2
**Presentation:** 3
**Contribution:** 3
**Rating:** 5
**Confidence:** 4

**Summary:**

The paper proposes MARIN, a novel method integrating implicit neural representations (INRs) with meta-learning to reconstruct 4D ocean temperature fields from sparse in-situ observations. The method addresses key challenges in ocean measurements, such as uncertainty estimations and data sparsity, by combining continuous implicit neural representations with meta-learning strategies.

**Strengths:**

High utility, relevance, and novelty.

Incorporates uncertainty estimates to handle measurement errors.

**Weaknesses:**

The results are preliminary, not a lot of validation is there. The analysis is not very detailed.

Time averages are good, but it would be good to see how differences manifest in instantaneous fields. This is a key problem with ocean or atmospheric mesoscales, too; RMSE does not always capture the small scales well.

Lack of clarity in the detailed methodology, particularly in the implementation of meta-learning and uncertainty estimation.

How about statistically downscaling (with the aid of climate models) to get provide accurate measurements over smaller grid scales than sampled?

Reproducibility issues- I don't find clarity in terms of model parameters and training, they have discussed for SIREN though.

**Questions:**

How well MARIN could replicate the Kinetic energy spectrum of the atmosphere  - and compare it with previous estimates. KE spectrum is an important quantity to measure and get right. Replicating mean is easy.

More details on the implementation of meta-learning and uncertainty estimation?

Ocean mesoscales drive a majority of the heat transport, so the small-scale differences in atmospheric fields might be important. In the same spirit, can authors really replicate a float trajectory and get similar measurements?

How does the proposed meta-learning framework specifically contribute to the improvements in reconstruction accuracy?

How about statistically downscaling (with the aid of climate models) to get provide accurate measurements over smaller grid scales than sampled?

---

### Official Review · Reviewer_goay · 2024-11-04

**Soundness:** 3
**Presentation:** 2
**Contribution:** 2
**Rating:** 6
**Confidence:** 3

**Summary:**

This article makes use of Implicit Neural Representation to model thd
temperature field of the ocean in 4 dimensions (latitude, longitude,
depth and time). The main contribution is to provide a fine-grained
model at the temporal scale (1 day instead of 1 month in prior works)
from only sparse measurement (that is real data available and not the
output of interpolation products) along with a prediction ability. With
the help of a few-shot learning technique, and contrary to usual INR
models, the representation is not only viable for a given period of
time, but also for future days. An experimental validation compares the
proposed method with traditional physical models.

**Strengths:**

A few existing techniques (INR, few-shot, uncertainty modeling) are
combined in a novel way for the modeling of the ocean dynamic. The most
interesting point for me is the combination of INR and few-shot learning
to give the model a predictive ability. This overcome a limitation of
INR which is to model only the state at a given time. Experiments are
convincing from the applied point of view but may be seen as limited for
a machine learning perspective. The overall structure of the article is
good enough.

**Weaknesses:**

The main problem of this article is the writing. Some parts are barely
understandable. Section 3.1 is particularly problematic from this point
of view: it is not clear if the training is made for 1 day or for all
the period, function names seems to change ($f$ and $g$). The meaning
of | changes from Equation (1) to Equation (2) (given some points, then
given a loss). The support of sums Σ is not clear and change in
unpredictable way (it is not enough to reuse letters from the previous
lines to given an insight of the source of the terms). Another
problematic section is the Algorithm 1: the term "Projection function"
was not introduced before, the bold expression "Inner loop" and "Outer
loop" are mislead since it qualifies the while loop and the for loop and
not the inside of the loop. About the uncertainty estimation, it's
confusing to see that in some parts the prediction target is a
scalar and in another parts a distribution.

I have the feeling that this work may not be adequately targeted to the
machine learning community.

Misc remarks
- lines 072 073: no idea of the meaning of tildas
- subsection 3.2.3: make clear that this works for a fixed theta; "while
  the outer-loop" a word may be missing; "meat-learner"
- the acronym OI should be developed everywhere, since it is not space
  consuming and not commonly known in the ML community
- line 327 "generalizes to unseen data" should make clear you are
  speaking of future days
- All figures should be black and white proof and color-blind proof
- Figure 2 is barely readable
- lines 411-413: use of "will" is surprising

**Questions:**

How is the uncertainty prediction evaluated ? I am usually not convinced
by this kind of output. Is it supposed to model the uncertainty from the
observations or the uncertainty of the prediction ?

Unlimited resolution provided by INR is interesting in theory, but I do
not see how to evaluate in practice. Figure 2 shows a non-local
evaluation, but only with a broad view. Are you able to evaluate the
quality at a sub-degree scale ? Or maybe to reconstruct some kind of
fine grained structures (like eddies for example) ?

Figure 1 show a shift of the error peak obtained by MARIN, could you
comment on this ?

What about the prediction time of the model ? It's obviously more
compact than a big tensor storing all the field, but the tensor gives an
answer instantly. The network is relatively small so the prediction time
should not be a problem, right ?

---

### Official Review · Reviewer_UBcv · 2024-11-12

**Soundness:** 2
**Presentation:** 2
**Contribution:** 2
**Rating:** 3
**Confidence:** 4

**Summary:**

The authors use implicit neural representation (INR) to represent physical fields describing the dynamics of the evolution of ocean properties. The INR architecture is used, which simultaneously predicts both the mean and the variance to simulate uncertainty. The meta-learning approach based on fine-tuning the model using a dedicated test sample further improves the accuracy of the forecast. Based on a number of real data describing the properties of the ocean, the proposed method showed high accuracy of the forecast.

**Strengths:**

1) the authors considered the problem statement, related to the restoration of the values of the physical field given partial measurements, which is important for practical applications

2) the authors demonstrated that INR, in the case of processing data on various characteristics of the ocean, makes it possible to achieve a high quality forecast

3) the authors conducted a fairly detailed experimental analysis of the properties of the proposed method based on data on ocean characteristics

**Weaknesses:**

1) This work is not the first in which INК is used to model a physical system, fields of physical properties, see, for example,

Continuous PDE Dynamics Forecasting with Implicit Neural Representations, ICLR 2023
https://openreview.net/forum?id=B73niNjbPs

Continuous Field Reconstruction from Sparse Observations with Implicit Neural Networks, ICLR 2024
https://openreview.net/forum?id=kuTZMZdCPZ

Therefore, from this point of view, this work is more application-driven rather than method-driven in the sense that the authors show how the existing INR-based approach with some additional modifications works when modeling fields of physical properties of the ocean.

2) the description of the architecture of the model and the protocol for conducting experiments is not very detailed: for example, a) it is not clear how the dynamics of physical properties between observations at different moments in time is taken into account; b) the architecture of the branch of the neural network that predicts uncertainty is not clear; what benefit does this additional forecast bring? c) it seems that not all necessary baseline methods for comparison were considered - for example, a comparison with reconstruction methods based on tensor decomposition was not provided.

**Questions:**

1) Using INR the authors trained a separate model for each month (12 models in total). What is about the continuity of the transition from model to model between months? How to take dynamics w.r.t. time into account when constructing a model even for one month? How to guarantee smoothness of transitions between consecutive days? Neural Networks can have spurious oscillations so that transition can be not always physically plausible.

2) How to evaluate quality of uncertainty estimate? Why do we need it? The authors did not provide any evidence of efficiency of the proposed uncertainty estimate. Moreover, if sigma is modelled is a separate NN branch which depends on input vector x, during training sigma can converge to zero, so the overall criterion in (5) will be not stable.

"Our approach here not only allows the model to represent the ocean field data but also provides data-driven uncertainty
estimates, addressing limitation of INRs in this context." - Please, provide some evidence that these data-driven uncertainty
estimates are useful in practice.

3) 215-216: "The inner-loop samples and optimises daily profiles on N days, while the outer-loop." - the sentence is not finished.

4) 253:  "we train a separate model for each month (twelve models in total)." - this looks like too computationally intensive. Is it possible somehow to re-use results from previous months/days? Also, I propose to specify computational complexity of the proposed method and compare it with computational complexities of competing approaches.

5) It is not clear why the authors did not incorporate any dynamics w.r.t. time when fitting the model to data inside the considered month.

6) The authors proved that they can represent ocean physical fields using INR. How does efficiency of such representation influence any further downstream task? Is it possible to demonstrate effectiveness of the proposed approach based on some downstream task?

---

### Author Response · Authors · 2024-11-21
**General Response**

We sincerely thank all reviewers for their time and thoughtful comments on our submission. It is interesting to receive comments from five reviewers from diverse perspectives and we are excited to address the questions and suggestions raised. Below, we summarize some general points before addressing specific feedback in detail.

**Motivation and Goal.**
The primary goal of our work is to construct global observational ocean products from raw oceanic observations. Specifically, we aim to estimate the ocean state on a daily basis using observations available for a given day. Our focus is on achieving high accuracy in reconstructing daily ocean fields from these observations. To clarify, our work is most appropriately compared with ocean-gridded products derived through optimal interpolation methods, where we used INRs. We recommend referring to [1,2] for a clear understanding of this task, and how our approach differs from PDE learning and data assimilation tasks.

To the best of our knowledge, no previous work has explored the use of AI methods to directly construct daily ocean temperature fields from raw observational data. This makes our work a novel contribution that we hope will inspire broader attention from the AI community. Evaluating the quality of such reconstructions poses unique challenges. Currently, we use RMSE as a metric, computed against high-resolution data and point-like observational data, where the latter provides location precision unmatched by existing grids.

We acknowledge and appreciate the reviewers' suggestions for improving evaluation metrics. For example, Reviewer Goay suggested incorporating metrics for unlimited resolution, Reviewer Kzma proposed evaluating instantaneous fields, and Reviewer Z9NN highlighted the inclusion of confidence intervals. These are valuable insights, and while we aim to incorporate them in future work, we note that their adaptation to an ML context may require additional development.

**Uncertainty.**
Predicting uncertainties is proposed here in alignment with traditional observational gridded products using optimal interpolation methods. This prediction uncertainty has been used in climate science, for example,

ClimODE: Climate and Weather Forecasting with Physics-informed Neural ODEs: https://openreview.net/forum?id=xuY33XhEGR

**Dynamics, Forecasting, and Methodological Improvements.**
Several reviewers have suggested exploring ocean dynamics, forecasting capabilities, and methodological enhancements. While these are indeed important and intriguing directions, our current focus is on demonstrating the accuracy and viability of AI-generated ocean products. Establishing this foundation is a critical step before extending our work to such applications.

**References**

1.	Chunling Zhang, Danyang Wang, Zenghong Liu, Shaolei Lu, Chaohui Sun, Yongliang Wei, and Mingxing Zhang. Global gridded argo dataset based on gradient-dependent optimal interpolation. Journal of Marine Science and Engineering, 10(5):650, 2022.

2.	Hong Li, Fanghua Xu, Wei Zhou, Dongxiao Wang, Jonathon S Wright, Zenghong Liu, and Yanluan Lin. Development of a global gridded argo data set with barnes successive corrections. Journal of Geophysical Research: Oceans, 122(2):866–889, 2017.

---

> ### Comment · Reviewer_goay · 2024-11-25
>
> Thanks for your response. I see no reason to changes my marks.

---

> ### Comment · Area_Chair_ZMiQ · 2024-11-26
>
> Dear reviewers,
>
> Please make to sure to read, at least acknowledge, and possibly further discuss the authors' responses to your comments. Update or maintain your score as you see fit.
>
> The AC.

---

> ### Comment · Reviewer_Z9NN · 2024-11-26
> **Answer to the authors**
>
> Thank you for your response. I still find that the submission lacks a technical comparison with other state of the art optimal interpolation approaches, and that there are a number of clarity issues in the presented methodology. Therefore I maintain my score.

---

> ### Comment · Reviewer_5FcN · 2024-11-26
>
> I have reviewed the authors' responses, which clarified some minor points, such as the motivation behind the work and the treatment of uncertainty. However, I am disappointed that the authors did not adequately address the novelty of the proposed model. At present, the approach appears to involve running three baselines and selecting the best-performing one, raising concerns about its broader applicability to other climate datasets.
>
> If the claimed novelty lies in constructing daily ocean temperature fields from raw observational data, further evidence is needed. Specifically, the authors should demonstrate how the proposed method and the baselines handle key challenges, such as data sparsity. Additionally, the measurement distribution likely exhibits specific patterns informed by prior knowledge. The authors should explore how to incorporate this prior knowledge into their baseline models to enhance performance.
>
> Regarding uncertainty estimation, the evaluation remains insufficient. Incorporating Gaussian process-based baselines would provide a meaningful comparison for assessing the quality of uncertainty estimates. Relevant approaches include SVGP [1], DGP [2], and GPSat [3].
>
> Finally, I am not convinced by the effectiveness of the proposed meta-learning approach. A simpler adaptation strategy, such as daily fine-tuning without meta-learning, might achieve comparable results. Furthermore, the manuscript overlooks several existing methods for adapting implicit neural representations (INR) to large-scale fields, such as context-pruned meta-learning [4] or meta-SGD [5]. Including and comparing against these methods would provide a more comprehensive evaluation of the proposed approach.
>
> References:
> [1] James Hensman, Nicolo Fusi, and Neil D Lawrence. Gaussian processes for big data. Uncertainty in Artificial Intelligence, 2013.
> [2] Hugh Salimbeni and Marc Deisenroth. Doubly stochastic variational inference for deep Gaussian processes. Advances in Neural Information Processing Systems, 2017.
> [3] William Gregory, Ronald MacEachern, So Takao, Isobel R Lawrence, Carmen Nab, Marc Peter Deisenroth, and Michel Tsamados. Scalable interpolation of satellite altimetry data with probabilistic machine learning. Nature Communications, 2024.
> [4] Tack, J., Kim, S., Yu, S., Lee, J., Shin, J., and Schwarz, J.R., 2024. Learning large-scale neural fields via context-pruned meta-learning. Advances in Neural Information Processing Systems, 36.
> [5] Li, Z., Zhou, F., Chen, F., and Li, H., 2017. Meta-SGD: Learning to learn quickly for few-shot learning. arXiv preprint arXiv:1707.09835.

---

### Comment · Area_Chair_ZMiQ · 2024-11-26

Dear all,

The deadline for the authors-reviewers phase is approaching (December 2).

@For reviewers, please read, acknowledge and possibly further discuss the authors' responses to your comments. While decisions do not need to be made at this stage, please make sure to reevaluate your score in light of the authors' responses and of the discussion.

- You can increase your score if you feel that the authors have addressed your concerns and the paper is now stronger.
- You can decrease your score if you have new concerns that have not been addressed by the authors.
- You can keep your score if you feel that the authors have not addressed your concerns or that remaining concerns are critical.

Importantly, you are not expected to update your score. Nevertheless, to reach fair and informed decisions, you should make sure that your score reflects the quality of the paper as you see it now. Your review (either positive or negative) should be based on factual arguments rather than opinions. In particular, if the authors have successfully answered most of your initial concerns, your score should reflect this, as it otherwise means that your initial score was not entirely grounded by the arguments you provided in your review. Ponder whether the paper makes valuable scientific contributions from which the ICLR community could benefit, over subjective preferences or unreasonable expectations.

@For authors, please respond to remaining concerns and questions raised by the reviewers. Make sure to provide short and clear answers. If needed, you can also update the PDF of the paper to reflect changes in the text. Please note however that reviewers are not expected to re-review the paper, so your response should ideally be self-contained.

The AC.

---

### Note · Authors · 2024-12-03

**Comment:**

According to the review from the reviewers, we receive some good comments. However, based on the problem we defined, the evaluation suggestions here require enough physical background which we do not think is a good fit for the AI community. Because of this, we decide to withdraw our submission.

**Withdrawal Confirmation:**

I have read and agree with the venue's withdrawal policy on behalf of myself and my co-authors.